# Hybrid Silver-Containing Materials Based on Various Forms of Bacterial Cellulose: Synthesis, Structure, and Biological Activity

**DOI:** 10.3390/ijms24087667

**Published:** 2023-04-21

**Authors:** Alexander Vasil’kov, Ivan Butenko, Alexander Naumkin, Anastasiia Voronova, Alexandre Golub, Mikhail Buzin, Eleonora Shtykova, Vladimir Volkov, Vera Sadykova

**Affiliations:** 1A.N. Nesmeyanov Institute of Organoelement Compounds, RAS, 119334 Moscow, Russia; solmaersgold@gmail.com (I.B.); naumkin@ineos.ac.ru (A.N.); voronova.anastasiia.a@mail.ru (A.V.); golub@ineos.ac.ru (A.G.); buzin@ineos.ac.ru (M.B.); 2G.F. Gause Institute of New Antibiotics, 119021 Moscow, Russia; sadykova_09@mail.ru; 3Shubnikov Institute of Crystallography, FSRC “Crystallography and Photonics” RAS, 119333 Moscow, Russia; eleonora.shtykova@gmail.com (E.S.); vvo@ns.crys.ras.ru (V.V.)

**Keywords:** hybrid materials, metal–vapor synthesis, silver nanoparticles, bacterial cellulose, X-ray photoelectron spectroscopy, powder X-ray diffraction, small-angle X-ray scattering, antimicrobial activity

## Abstract

Sustained interest in the use of renewable resources for the production of medical materials has stimulated research on bacterial cellulose (BC) and nanocomposites based on it. New Ag-containing nanocomposites were obtained by modifying various forms of BC with Ag nanoparticles prepared by metal–vapor synthesis (MVS). Bacterial cellulose was obtained in the form of films (BCF) and spherical BC beads (SBCB) by the *Gluconacetobacter hansenii* GH-1/2008 strain under static and dynamic conditions. The Ag nanoparticles synthesized in 2-propanol were incorporated into the polymer matrix using metal-containing organosol. MVS is based on the interaction of extremely reactive atomic metals formed by evaporation in vacuum at a pressure of 10^−2^ Pa with organic substances during their co-condensation on the cooled walls of a reaction vessel. The composition, structure, and electronic state of the metal in the materials were characterized by transmission and scanning electron microscopy (TEM, SEM), powder X-ray diffraction (XRD), small-angle X-ray scattering (SAXS) and X-ray photoelectron spectroscopy (XPS). Since antimicrobial activity is largely determined by the surface composition, much attention was paid to studying its properties by XPS, a surface-sensitive method, at a sampling depth about 10 nm. C 1s and O 1s spectra were analyzed self-consistently. XPS C 1s spectra of the original and Ag-containing celluloses showed an increase in the intensity of the C-C/C-H groups in the latter, which are associated with carbon shell surrounding metal in Ag nanoparticles (Ag NPs). The size effect observed in Ag 3d spectra evidenced on a large proportion of silver nanoparticles with a size of less than 3 nm in the near-surface region. Ag NPs in the BC films and spherical beads were mainly in the zerovalent state. BC-based nanocomposites with Ag nanoparticles exhibited antimicrobial activity against *Bacillus subtilis*, *Staphylococcus aureus*, *Escherichia coli* bacteria and *Candida albicans* and *Aspergillus niger* fungi. It was found that AgNPs/SBCB nanocomposites are more active than Ag NPs/BCF samples, especially against *Candida albicans* and *Aspergillus niger* fungi. These results increase the possibility of their medical application.

## 1. Introduction

Bacterial cellulose is a pure natural biomaterial generated by some aerobic bacteria has a wide range of useful properties: nanofibrillar microporous structure, gas permeability, high mechanical strength, good water-absorption capacity, homogeneous structure, biocompatibility, biodegradability, and extreme hydrophilicity [1,2,3,4]. Due to these properties, various forms of cellulose have found application in the food industry [5,6], as personal hygiene products (face masks) [7], and medicine (wound dressing, artificial skin, drug delivery, hemostatic materials, vascular grafts, skeletons for tissue engineering, biosensors, and diagnostics) [8,9,10,11,12].

To obtain bacterial cellulose, two methods of culturing cellulose-synthesizing bacteria are used—stationary and dynamic [13]. With the stationary method, bacterial cellulose films are obtained [14]: under conditions of dynamic cultivation, the polymer is formed in the form of spherical bacterial cellulose beads of various diameters [4]. With the first method, a BC film is formed at the air–liquid interface of the culture medium. With the second one, small irregularly shaped BC granules are formed throughout the entire volume of the nutrient medium. BC spheres, unlike films, have lower mechanical strength and degree of polymerization than BC film. At the same time, the morphology of spherical BC is characterized by an increased surface area, a more compacted three-dimensional mesh structure, and a high release rate of loaded active substances [4,15]. The method of obtaining different forms of BC is determined by the further scope of its application [16]. The main disadvantage of the stationary cultivation method is the high cost of production due to the low productivity of the process [14]. The most technologically advanced for industrial production is the dynamic method carried out in fermenters [17].

The lack of biological activity in BC leads to the ineffectiveness of its use as a dressing agent, since such a material cannot prevent infection of the wound and does not contribute to its treatment [18]. The BC surface contains hydroxyl groups, which makes it possible to modify it with chitosan [19], benzalkonium chloride [20], silver sulfadiazine [21], nanoparticles of biologically active metals, and others. This removes restrictions on the use of BC in the treatment of wounds and avoids secondary infection [22]. In this regard, the development of new environmentally friendly methods for modifying BC with biologically active components is an urgent and actively developing direction for the creation of medical materials.

BC modification can be carried out *in situ* and *ex situ*. *In situ* modification consists in changing BC properties during cultivation by controlling the conditions of this process and adding compounds to the culture, including those that change the structure of the future polymer, for example, carboxymethylcellulose and sodium fluoride [23,24]. *Ex situ* modification is carried out after the BC has been formed, and is carried out by either chemical or physical methods [23].

Currently, one of the intensively developed scientific directions is the use of metal nanoparticles with biological activity to create medical materials. Silver nanoparticles have long attracted the attention of researchers from the field of medicine and health care due to concerns about the resistance of bacteria to antibiotics. Ag NPs interact with compounds contained in the respiratory enzymes of bacteria, which leads to inhibition of their respiration and death. Compared to other traditionally used sterilizing agents, Ag NPs have high temperature stability [25]. They exhibit antibacterial activity with an effective broad spectrum of action against various strains of aerobes, anaerobes, and Gram-positive and Gram-negative microorganisms resistant to antibiotics [26]. A significant advantage of Ag-containing compounds and Ag NPs over other microbiological agents is their increased toxicity to microorganisms with lower toxicity to mammalian cells [27]. Numerous studies have shown that a bacterial cellulose composite with Ag has antibacterial activity against both Gram-positive and Gram-negative bacteria [28].

Modification of BC with biologically active metal nanoparticles expands the possibilities of its use for various biomedical applications [29,30,31]. There are several methods for obtaining and introducing metal nanoparticles into the bacterial cellulose matrix [21,32]. Since BC contains a significant amount of hydroxyl groups, Ag NPs can be chemisorbed by the surface of BC nanofibers. For the synthesis of metal nanoparticles in biopolymer matrices, as a rule, the procedure of chemical reduction of metal salts is used [33]. The dependence of the size and shape of the synthesized Ag NPs, as well as the antibacterial efficiency of Ag-containing BC composites on the reaction conditions, have been established [34].

Methods for the reduction of metals using sodium borohydride [35], tannic acid [36], triethanolamine [37] and ultraviolet radiation [12] have a number of significant limitations: a long synthesis process, the presence of a significant amount of impurities in the form of surfactants and residues of synthesis products, as well as the complexity of controlling the completeness of metal reduction [38,39].

Metal–vapor synthesis (MVS), an effective environmentally friendly method for producing biologically active metal nanoparticles and their oxides, is widely used in the preparation of new biomedical hybrid metallopolymers [40,41]. MVS is based on simultaneous processes of evaporation and condensation of metal and organic ligand on the reactor walls cooled by liquid nitrogen under vacuum conditions of 10^−4^–10^−6^ Torr. There are no restrictions when choosing a metal or a combination of metals, and it can be carried out for almost any combination of organic reagent and metal. MVS is a technologically advanced closed cycle that can be effectively integrated into various processes for production of composites containing nanoparticles of Au, Ag, Cu, and other metals [42,43,44,45]. The particle size is controlled by varying the nature of the metal–organic reagent pair and some other synthesis parameters (pressure, evaporation rate, etc.). The advantage of this method for obtaining metal nanoparticles is the absence of synthesis by-products during their formation, which is especially important for the purity of biomedical materials.

Bacterial cellulose has different morphology, surface composition, crystallinity and other characteristics depending on the method of cultivation—stationary or dynamic. The aim of this work was to compare the composition, structure and biological properties of silver-containing nanocomposites derived from two forms of BC and the prospects for their use as biologically active systems. Despite the significant amount of existing work on the synthesis and study of metal-containing nanocomposites based on bacterial cellulose, this area of research is practically not presented in the literature, although it is the change in BC surface composition that can significantly affect the stabilization of silver nanoparticles on its surface and the properties of the hybrid material as a whole.

## 2. Results and Discussion

In the process of stationary and dynamic cultivation, BC materials were obtained as films and spheres, respectively, and were modified with organosol Ag–isopropanol synthesized by MVS. Therefore, two types of composite—Ag NPs/BCF and Ag NPs/SBCB—were obtained. The scheme for obtaining Ag-containing materials is shown in Figure 1. Appendix A shows the initial forms of BC (a) and SBCB (c) and the silver-containing forms Ag NPs/BCF (b) and Ag NPs/SBCB (d).

The spatial organization of the structure of the two forms of BC was studied by SEM. Figure 2 shows the SEM micrographs of BCF and SBCB.

As a rule, bacterial cellulose obtained under conditions of liquid-phase cultivation is a cellular structure consisting of layers located parallel to the surface. The BC consists of filamentous fibers combined into microfibrils with a diameter of 15–30 nm. Microfibrils are combined into macrofibrils, which are present in the structure of any BC form with a diameter of 60–120 nm. Appendix A shows histograms of the size distribution of fibrils for two forms of BC.

It was shown [46,47] that during cellulose cultivation, thin primary layers are first formed and then new thicker secondary layers are formed on top, the density of which is not uniform due to a decrease in the concentration of the carbon source over time [48]. It was found that primary layers with a diameter of about 1.3 μm are present in the structures of the obtained BCF, which are at a distance of 4–5 μm from each other, forming areas with a low density of fibrils. The distance between the primary layers is about 4.5 μm, which is in good agreement with the previously established data [49,50]. Analysis of BCF micrographs showed that there are secondary layers with a thickness of about 13 μm and a distance between them of 12.5 μm. The density of the secondary layers is not uniform—areas with both high and low density are recorded.

In the case of SBCB, primary layers with a thickness of about 2.8 μm and a distance between the layers of 9.6 μm were also present. The secondary layers separated by a distance of 5.6 μm have a thickness of 8.3 μm. Analysis of the SEM data showed that the primary layers of BSF are thinner than those of SBCB [51], while the secondary layers are thicker in BCF, which characterizes the difference in the rate of formation of layers in static and dynamic cultivation. A similar trend was observed earlier, which is related to the rate of glucose conversion to BC and lower under dynamic conditions due to hydrodynamic stress inducing the accumulation of self-defense metabolites.

The preparation procedure of Ag-containing bacterial cellulose by MVS can be formally divided into three steps (Figure 3).

In the first step, the colloidal solution of nanoparticles in an organic solvent referred to as organosol (Ag–2-propanol) was prepared. The second step includes the impregnation of bacterial cellulose with the organosol. In the third step, the solvent is removed and the Ag-containing celluloses are dried in vacuo.

Organosol Ag–isopropanol was investigated by TEM. Previously, this technique was used to study peptidic “nanomillipede” [52]. Figure 4 shows a micrograph of Ag NPs and a histogram of the particle size distribution. Analysis of microphotographs showed that the particle size varies in the range of 1.7–5.6 nm with an average size of 2.8 ± 0.1 nm.

The analysis of BCF and SBCB composites modified by Ag NPs, carried out by SEM, did not reveal significant differences in the structure of Ag-containing systems for either Ag NPs/BCF or Ag NPs/SBCB. Basically, Ag NPs were located on the surface of BC fibrils in the form of small aggregates, similar to “bunches of grapes,” consisting of smaller nanoparticles (Figure 5).

Analysis of the morphology of the surface of the Ag NPs/BCF showed that the porous microfibrillar structure of bacterial cellulose is preserved after its modification by metal organosol, and Ag nanoparticles are fairly evenly distributed over the entire surface of the film, decorating BC nanofibrils. The bacterial cellulose nanocomposite is a gradient material in which the metal concentration decreased from the surface to the middle of the film.

Figure 6 shows the distribution maps of C, O, and Ag and the energy-dispersive X-ray spectra of the BCP and SBCB nanocomposites with Ag NPs. Elemental analysis in the EDX confirmed the presence of silver on the bacterial cellulose.

SEM EDX micrographs showed that Ag NPs deposited on the BC surface were located mainly in places with surface inhomogeneities—ridges, roughness, etc. Along with the elements C, O, and Ag, traces of Si and Al were observed, the sources of which are the aluminum table used when taking samples, as well as the adhesive tape fixing the samples.

To understand surface chemical composition of modified bacterial cellulose, XPS is widely used [53,54,55,56,57,58,59,60,61,62]. We analyzed and compiled literature data related to the C 1s, O 1s and Ag 3d_5/2_ spectra of BCF, Ag NPs/BCF, SBCB and Ag NPs/SBCB and Ag NPs (Table 1 and Table 2).

The C 1s and O 1s photoelectron spectra of the reference sample of cellulose are characterized by components at 286.73, 288.06, 532.93 and 533.51 eV related to C-OH and O-C-O groups [63]. The intensity ratios of C-OH and O-C-O components in the C 1s and O 1s spectra are 5:1 and 3:2, respectively. At the same time, the photoelectron C 1s and O 1s spectra of non-reference samples are characterized by additional components, which are mainly due to the presence of C-C/C-H and C(O)O groups. Their presence is attributed to adventitious carbon, defects due to a chemical change in the cellulose structure, contaminations including an air-exposed surfaces, and inhomogeneous charging [64,65,66]. Thus, in the spectra of C 1s as a rule, three or four components with binding energies of ~285.0 (C1), 286.73 (C2), ~288.1 (C3) and ~288.6 eV (C4) are attributed to C-C/C-H, C-OH, O-C-O and C(O)O groups, respectively [55,56,57,58,59,62,64,65,67,68,69,70,71]. The literature data given in Table 1 are in reference to the C2 component at 286.73 eV. Along with the binding energies, they contain the relative intensities of the chemical groups. The binding energies of the C1 and C4 peaks are in the ranges 284.93–285.43 eV and 288.56–289.73 eV, respectively. The relative intensities of these peaks are in the ranges of 5–29% and 5.05–12%, respectively. Toro et al. noted that carboxylic groups can promote reduction and stabilization of well-dispersed Ag nanoparticles [72]. The component at 283.93 eV (C0) was assigned to low molar mass species.

**Table 1 ijms-24-07667-t001:** Characteristics of the C 1s photoelectron spectra: binding energies (E_b_) and relative intensities (I_rel_) of photoelectron peaks assigned to different chemical groups.

Sample		C-C/C-H *	C-C/C-H	C-OH	O-C-O	C(O)O	C=O	Ref.
	Peak	C0	C1	C2	C3	C4	C5	
Cellulose	E_b,_ eV		285.06	286.73	288.02	289.1		[64]
	I_rel_		7.22	64.56	23.15	5.05		
Cellulose	E_b,_ eV	283.93	285.43	286.73	287.73	289.13		[65]
	I_rel_	5.0	20.7	37.2	27.4	9.6		
Ag–cellulose	E_b,_ eV	283.93	285.43	286.73	287.73	289.13		[65]
	I_rel_	4.1	21.6	31.4	28.8	14.1		
Cellulose	E_b,_ eV		284.93	286.73	288.23			[54]
	I_rel_		16.0	62.6	21.4			
Cellulose	E_b,_ eV		285.03	286.73	288.08	288.56		[55]
	I_rel_		5	75	15	5		
Ag–cellulose	E_b,_ eV		285.4	286.73	288.06	289.37	288.06	[55]
	I_rel_		54	26	5	7	8	
Cellulose	I_rel_		285.23	286.73	288.23			[67]
Cellulose	I_rel_		285.09	286.73	288.34			[68]
Cellulose	E_b,_ eV		285.03	286.73	288.23	289.33		[69]
	I_rel_			80	20			
Cellulose			285.03	286.73	288.23	289.28		[70]
			16.87	65.62	15.65	1.86		
Cellulose	E_b,_ eV		285.03	286.73	288.13	288.93		[57]
Cellulose	E_b,_ eV		285.13	286.73	288.13	289.73		[58]
	I_rel_		6	70	23	1		
Cellulose	E_b,_ eV		285.14	286.73	288.02	288.96		[56]
	I_rel_		29	50	10	12		
Ag–cellulose	E_b,_ eV		285.11	286.73	288.03	288.95		[56]
	I_rel_		28	52	10	10		
Ag–cellulose	E_b,_ eV		285.23	286.73	288.13	289.23		[59]
	I_rel_		77	13	3	7		
Cellulose	E_b,_ eV		284.83	286.73	287.83	288.83		[62]
Cellulose	E_b,_ eV		285.03	286.73	288.23	289.23		[71]

*—low molar mass species.

The O 1s spectra also have additional components at ~532.1 and 534.3 eV related to non-equivalent C(O*)O and C(O)O* atoms in the C(O)O group, which are distinguished in the O 1s spectra. The corresponding literature data are listed in Table 2. In the case of the presence of metal nanoparticles, such as silver, a component at ~529.6 eV characteristic of the Ag-O bond is added [73,74]. However, since the concentration of silver is very low, the isolation of this component is difficult. It should be noted that binding energy of 531.13 eV [59], which is rather different from 529.6, indicates difference in oxide species.

**Table 2 ijms-24-07667-t002:** Characteristics of the O 1s photoelectron spectra: binding energies (E_b_) and relative intensities (I_rel_) of photoelectron peaks assigned to different chemical groups.

Sample		Ag-O	C(O*)O	C-OH	O-C-O	C(O)O*	Ref
	Peak		O1	O2	O3	O4	
Cellulose	E_b,_ eV		531.18	533.18	533.85		[64]
	I_rel._		2.96	73.45	23.58		
Cellulose	E_b,_ eV		532.1	532.8	533.4	534.0	[55]
	I_rel._		10	48	32	10	
Ag–cellulose	E_b,_ eV		532.0	532.7	533.4	534.1	[55]
	I_rel._		14	39	26	14	
Cellulose	E_b,_ eV			532.93	533.53		[69]
	I_rel._			0.6	0.4		
Ag–cellulose	E_b,_ eV	531.13	532.23	532.83	532.93	534.23	[59]
	I_rel._	18	18	9	37	18	
Cellulose	E_b,_ eV			532.93	533.51		[70]
	I_rel._			78.18	21.82		

Figure 7 displays the C 1s spectra fitted with four Gaussian peaks at ~285.0 (C1), 286.73 (C2), ~288.1 (C3) and ~288.6 eV (C4) attributed to C-C/C-H, C-OH, O-C-O and C(O)O groups, respectively. When the spectra were fitted with several Gaussian profiles, the relative intensities of the peaks at 286.73 and ~288.09 eV, characteristic of cellulose, retained the ratio 5:1, corresponding to the chemical structural formula of cellulose. As a reference, the C 1s spectrum of silver black obtained from the Ag–isopropanol organosol is presented, with peaks of 284.89 eV (C-C/C-H), 286.45 eV (C-OH), 287.94 eV (O-C-O) and 289.99 eV (C(O)O). The characteristics of the photoelectronic spectra are given in Table 1.

A comparative analysis of the C 1s spectra of BCF and SBCB shows that their compositions differ significantly. The content of C-C/C-H groups in SBCB is 2.6 times higher than in the film. Such a difference, other conditions being equal, may be due to biochemical processes that occur during dynamic cultivation [4]. A similar significant change in the surface compositions is observed after modification of both types of BC with Ag–isopropanol organosol. The content of the C-C/C-H group in SBCB after modification with Ag increased by threefold and in BCF 2.15-fold, which is associated with the presence of a carbon shell on Ag NPs formed during the synthesis of Ag nanoparticles. According to Figure 7 and Table 3, the relative intensity of C-C/C-H groups in Ag NPs is 77%. An increase in content of C-C/C-H groups is explained by the interaction of nanoparticles with isopropanol, which was used during MVS to stabilize Ag NPs [75]. An increase in the relative proportion of C-C/C-H groups was observed for all samples containing silver Ag (Table 1) excluding microcrystalline cellulose.

The O 1s spectra of BCF shown in Figure 8 are characterized by four peaks at 532.10, 532.82, 533.40, and 534.0 eV, which are assigned to C(O*)O, C-OH, O-C-O, and C(O)O*. When the spectra were fitted, for the functional groups C-OH and O-C-O, the intensity ratio of 3:2 was retained [63]. After the modification of BCF with Ag nanoparticles, two additional peaks at 530.60 and 531.65 eV were recorded, which were assigned to the Ag-O and C=O groups (Table 4) contained in Ag NPs, and a slight decrease in the intensity of carboxyl groups. A similar behavior is also observed for the modified SBCB; however, the proportion of carboxyl groups almost did not change.

As regards the binding energies of the Ag 3d _5/2_ photoelectronic peaks, according to our knowledge, the most accurate values for solid silver are 368.327 and 368.299 eV for Al Kα and Mg Kα, respectively [76]. The binding energies above and below these values should be assigned to the Ag^0^ state and Ag^+^ or Ag^2+^ states, respectively, excluding a case of Ag acetate [77,78].

Figure 9 shows the Ag 3d photoelectron spectra of composites and Ag NPs, and Table 5 lists their characteristics. The binding energies of the photoelectron peaks are close and indicate the presence of the Ag^0^ state, while the differences in their values indicate the appearance of a size effect [79,80]. The presence of plasmon loss peaks in the Ag 3d spectra of Ag NPs and celluloses at ~372 and 378 eV also indicates the presence of zerovalent silver [77]. The difference in the binding energies, relative intensities of the satellite peaks and the widths of the main peaks indicate the presence of another state of the Ag atoms, namely Ag^+^, while the Ag 3d spectrum of Ag NPs indicates only the Ag^0^ state.

XRD patterns of the studied materials are shown in Figure 10, Figure 11 and Figure 12. As is known from previous studies, there are several structural allomorphs of cellulose in nature, and their content depends on the origin of cellulose [81,82]. It was concluded that the dominative form of cellulose produced by bacteria is cellulose Iα [15,82]. The atomic structure of this allomorph was determined by Nishiyama et al. using the synchrotron X-ray and neutron diffraction [83]. The usage of the structural model of cellulose Iα (space group P1, Cambridge Structural Database code JINROO05) allowed the successful fit of the diffractograms for SBCB and BCF samples by the Rietveld method. In both cases, the simulated diffraction profiles are formed by numerous broadened, overlapped reflections, with the most intense ones contributing to the region of low 2θ angles. The strongest reflections are situated at 14.4° (010), 16.7° (001) and 22.5° (011) (Figure 10a and Figure 11a). It is worth noting that the intensity ratios between the reflections differs to some extent between SBCB and BCF, which can be explained by the different degree of preferred orientations of cellulose chains, inevitable for the cellulose materials [84]. The characteristic mean crystallite size of cellulose (D_c_) was calculated from the broadening of the peaks positioned within the 10–30° region using the Scherrer formula [85]. D_c_ values of 11 nm and 14 nm were obtained for SBCB and BCF, respectively. Cellulose crystallinity (CI) was evaluated by accounting for the contribution of amorphous structure to the overall pattern by the method previously described by Zhang et al., which implies modeling of this contribution by two halos centered at 20.5° and 38.9° [86]. This gives CI values of 85% for SBCB and 89% for BCF. Somewhat higher crystallinity and crystal size observed for the BCF cellulose, is in line with previous studies of related materials [15,87].

Figure 12 shows XRD patterns of Ag NPs obtained by MVS. Strong reflections are observed at 38.1°, 44.3°, 64.5°, 77.4° and 81.5°. Their positions and intensities correspond to, respectively, 111, 002, 022, 311, and 222 reflections of the Ag metal phase of cubic symmetry (space group Fm-3m, *a* = 4.087 Å) [88], as follows from the perfect fit of the pattern with this phase. The crystallite size of Ag nanoparticles of 21 nm was thus calculated by the abovementioned Scherrer line-broadening analysis.

The patterns of Ag NPs/SBCB and Ag NPs/BCF (Figure 10b and Figure 11b) clearly show the presence of both the cellulose and metallic silver. The phase of metal is evinced by its characteristic peaks. These peaks are observed at the same positions as with pure Ag nanoparticles, and the same abovementioned face-centered cubic phase of Ag metal is applicable for Rietveld-fitting the pattern above 35°. The calculated crystallite sizes for Ag (D_Ag_) amount to 21 nm (Ag NPs/SBCB) and 34 nm (Ag NPs/BCF). Comparison of these values with D_Ag_ of starting nanoparticles shows that no particle aggregation occurs during preparation of Ag NPs/SBCB, while it certainly takes place during preparation of Ag NPs/BCF.

The low-angle regions of PXRD patterns of composite materials (<35°) exhibit the same reflections as the patterns of corresponding initial pure celluloses. Calculation of the D_c_ values within the 10–30° interval, which is not affected by the metal additive, revealed that the cellulose crystallite size remains practically intact in the case of Ag NPs/BCF (14 nm), while it decreases from 11 to 7 nm in the case of Ag NPs/SBCB. This decrease in D_c_ means that the deposition process of Ag particles induces some perturbation in SBCB crystalline structure. The CI characteristic cannot be reliably calculated by the Zhang method for composite materials, because the reflections of Ag hinder correct determination of the area under amorphous halo positioned in the same region of diffraction angles. An assessment of crystallinity by the alternative empirical Segal method [89], which accounts for only the low-angle region, showed no prominent crystallinity drop upon deposition of metal nanoparticles, similar to the previously studied microcrystalline cellulose modified by Au using the MVS technique [90].

Small-angle X-ray scattering is a non-destructive technique allowing material structure determination in the range 1–250 nm with minimal sample preparation. The method is widely used to study size distributions of different nanoparticles, including metal ones, in bulk [91,92,93]. It provides statistically relevant information over a large volume (typically 1 mm^3^), thus being an ideal complement to microscopy techniques that provide only local information [94].

The experimental scattering profiles from the SBCB and Ag NPs/SBCB gels and lyophilized ones are presented in Figure 13a,b.

As one can see from Figure 13 all scattering curves are characteristic of systems with high polydispersity. The samples with incorporated silver have a higher scattering intensity due to the higher electron density of the metal, which is an indirect confirmation of the presence of silver in these specimens.

The volume size distributions *D_V_(R)* calculated using the GNOM program [95] for the gels and lyophilized gels as well as those with embedded Ag NPs are presented in the insets of Figure 13. The size distribution of inhomogeneities in SBCB is represented by one rather broad peak with an average *R* of about 2.5 nm. The sample also contains some amount of larger inhomogeneities with sizes up to 60 nm. After Ag NP incorporation into the gel, the *D_V_(R)* profile changes: the fraction of small inhomogeneities becomes narrower with an average *R* of about 1.5 nm, and a wide fraction of large scattering objects with sizes up to 80 nm appears. It can be assumed that the gel inhomogeneities are cavities in which the Ag NPs are localized during metal incorporation. In small cavities, correspondingly, small metal nanoparticles are stabilized, which subsequently merge into large aggregates. The soft nature of the gel does not prevent such aggregation, and most likely, metal nanoparticles can migrate quite freely in the gel volume.

Compared to SBCB gel, lyophilized SBCB gel initially exhibits a great number of large heterogeneities, which then serve as compartments for the formation of Ag NPs. The main fractions of Ag NPs incorporated into this sample are the narrow fraction of small particles with radii of 1.3 nm and a wide fraction of large particles with an average radius of 18.5 nm.

Despite the difference between the SBCB gel and its lyophilized variant, nanoparticles of almost the same size are formed in both samples.

Figure 14 and Figure 15 show the curves of thermogravimetric analysis (TGA) and differential thermogravimetry (DTG) of BCF, Ag NPs/BCF, SBCB, Ag NPs/SBCB and Ag NP samples.

Thermogravimetric analysis in air showed that the mass of Ag NPs is stable throughout the studied temperature range (Figure 14e). The initial cellulose, as well as its composite with Ag NPs, contained an insignificant amount of moisture (1–3%) sorbed from the air, which was removed from the samples in the temperature range from room temperature to 100 °C. Thermal oxidative degradation developed in the region of 320 °C with removal of hydrocarbons up to 550 °C (Figure 14a–d). With a further increase in temperature in the air, the mass of the sample did not change.

Thermal oxidative degradation of BCF and SBCB developed in the region of 326–320 °C (Figure 15a,c), which characterizes a slightly greater thermal stability of the BCF. It was found that the temperature of the maximum decomposition rate at this stage of destruction in Ag NPs/SBCB and Ag NPs/BCF begins at 333 °C and 326 °C, which is 7 °C and 6 °C higher than that of the original cellulose, respectively. The introduction of Ag NPs allowed a slight increase in the thermal stability of BCF and SBCB.

Preliminary tests of the obtained nanocomposites showed comparatively low inhibition zones against all test bacteria and fungi. The average zone of inhibition was 9 mm against all bacteria (Table 6).

Here, inhibition was found for the yeast *C. albicans* ATCC 2091, which was absent for the reference drug amphotericin B. Zones of inhibition were 7 ± 0.7 mm for Ag NPs/BCF and 10 ± 0.3 mm for Ag NPs/SBCB, respectively (Appendix A). However, the activity of the Ag NPs/SBCB films was higher than those of Ag NPs/BCF in all tested microorganisms. Many areas of human life experience a need for the use of biologically active materials containing silver nanoparticles. However, it should be taken into account that some of them can be toxic to humans and genotoxic to insects [96,97]. It has been established that these nanocomposites do not exhibit cytotoxic activity against mesenchymal stem cells [98].

Research reviews state that the strategies adopted in the design and development of BC composites with specific characteristics is achieved by integrating BC with specific nanoparticles. BC composites developed with such inorganic nanoparticles as Ag and Cu exhibit antimicrobial potential against several pathogens [99,100]. Nevertheless, not much information is available on the difference between the nanocomposites obtained from static and dynamic cultivation conditions. The different location of cellulose fibrils provides the activity of functional groups on the surface of nanocomposites that can be improved effectively in various types of cultivation. Studies have shown that Ag NPs/SBCB nanocomposites obtained with bacterial cellulose produced by dynamic cultivation are more active than nanocomposites with bacterial cellulose obtained by static cultivation.

## 3. Materials and Methods

### 3.1. Production of Bacterial Cellulose

Bacterial cellulose was obtained by culturing the Gluconacetobacter hansenii producer strain on GH-1/2008, which was carried out for 7 days at 28 °C under static and dynamic conditions on a modified medium Hestrin S. and M. Schramm: glucose (Thermo Fisher Scientific, Waltham, MA, USA)—40 g/L, yeast extract (Thermo Fisher Scientific, Waltham, MA, USA)—5 g/L, Na_2_HPO_4_ (AppliChem GmbH, Darmstadt, Germany)—2.7 g/L, K_2_HPO_4_ (AppliChem GmbH, Darmstadt, Germany)—2 g/L, (NH_4_)_2_SO_4_ (Sigma-Aldrich RTC, Inc., Laramie, WY, USA)—3 g/L, citric acid monohydrate (Thermo Fisher Scientific, Waltham, MA, USA)—1.15 g/L. The nutrient medium was sterilized in an autoclave (All American 50×, Wisconsin Aluminum, Manitowoc, WI, USA) at 120 °C for half an hour. At 22 °C, 5 mL of seed material and 1 mL of ethyl alcohol were introduced into the flasks with the medium. The introduction of the producer and alcohol into the flasks with media was carried out in a laminar flow cabinet (NEOTERIC BMB-II-”Laminar-C”-0.9б Industrial Group “Laborator”, St. Petersburg, Russia). To obtain BCF, flat-bottomed conical flasks with a volume of 750 mL with a medium of 200 mL were used, which were placed in a thermostat (RF 115, Binder GmbH, Tuttlingen, Germany). In the case of SBCB, spherical round-bottomed flasks were used (flask volume: 250 mL, pit media 125 mL), which were placed in a shaker incubator (KS 4000 control IC, IKA, Staufen, Germany) with a fixation of 120 rpm.

Purification of the polymer from the culture fluid and the producer was carried out using a RIPA buffer (Thermo Fisher Scientific, Waltham, MA, USA) with the sequential addition of the enzyme deoxyribonuclease I (Promega, Madison, WI, USA), and the procedure was repeated after 24 h. After that, different forms of BC were washed in distilled water, which was updated to obtain a neutral pH value of the flushing liquid.

### 3.2. Preparation of BCF and SBCB-Based Nanocomposites with Ag Nanoparticles

Silver nanoparticles were obtained by MVS in the setup described elsewhere [90]. Isopropanol (Fluka, Buchs, Switzerland, 99.8%) was used as an organic dispersion medium for the preparation of organosol Ag. Silver (99.99%) was evaporated by resistive heating from a tantalum boat. In a typical experiment, about 120 mL of organic reagent and 0.2 g of metal were used in the synthesis.

BCF were fixed on porous cylindrical stainless steel frames (5 × 7 cm^2^) and placed in a flask. The SBCB in isopropanol was placed in a flask with a stirring element. Both flasks in the process of feeding them organosol Ag were under a vacuum of 10 Pa. Modification was carried out in an argon atmosphere for 20 min with intensive stirring. Then, the excess organosol was removed from the flasks and the contents of the flasks dried to a constant mass in a vacuum of 10^−2^ Pa and a temperature of 60 °C.

### 3.3. Morphology of the Obtained Samples

TEM images were gained with a transmission electron microscope LEO 912AB OMEGA, Zeiss (Oberkochen, Germany) at an acceleration voltage of 100 kV. Particle size distribution histogram was obtained from electron micrographs. It was calculated by measuring the size of 200 displayed particles using the SigmaScan Pro software. The distribution was approximated with Gaussian function using SigmaPlot (11 version) (Systat Software Inc., Richmond, CA, USA). Scanning electron microscopy images for samples placed on a 25 mm aluminum table and secured with a conductive carbon tape were obtained in the secondary electron mode at an accelerating voltage of 15 kV and medium vacuum mode on a Hitachi TM4000Plus desktop electron microscope equipped with an energy dispersive X-ray spectrometer (QUANTAX 75, Bruker, Billerica, MA, USA).

### 3.4. X-ray Photoelectron Spectroscopy

The XPS analysis was performed using a Thermo Fisher Scientific Theta Probe (Thermo Fisher Scientific, Waltham, MA, USA) spectrometer. For analysis, a monochromatic Al Kα (1486.6 eV) X-ray source was used. The spectra were measured at room temperature at a pressure of ~5 × 10^−8^ Pa in the analytical chamber. The samples were mounted on a titanium sample holder with two-sided adhesive tape. The energy scale of the spectrometer was calibrated to provide the following values for reference samples (i.e., metal surfaces freshly cleaned by ion bombardment): Au 4f_7/2_–83.96 eV, Cu 2p_3/2_–932.62 eV, Ag 3d_5/2_–368.21 eV. Survey and high-resolution spectra of appropriate core levels were recorded at constant pass energies of 300 and 100 eV and with step sizes of 1 and 0.1 eV, respectively. The electrostatic charging effects were compensated by using an electron neutralizer. The binding energy scales for the cellulose samples were referenced to the C–OH component in the C 1s spectra at 286.73 eV [62] and for Ag black at 285.0 eV. After charge referencing, a Shirley-type background with inelastic losses was subtracted from the high-resolution spectra. The surface chemical composition was calculated using atomic sensitivity factors included in the software of the spectrometer corrected for the transfer function of the instrument.

### 3.5. Powder X-ray Diffraction

Powder X-ray diffraction phase analysis was performed with a D8 Advance (Bruker AXS, Karlsruhe, Germany) diffractometer in Bragg–Brentano focusing geometry using CuK_α_ radiation and an angular range of 5–90° with a step of 0.02° and scan rate of 0.5–2 deg min^−1^. The samples were placed on flat holders. Diffraction pattern profiles were fit using the TOPAS 5 program package (Bruker AXS, Karlsruhe, Germany).

### 3.6. Thermogravimetric Analysis

The thermal stability of silver nanoparticles, the initial BC and BC with Ag was studied by thermogravimetric analysis (TGA) and dynamic thermogravimetry (DTG) on the MOM, Hungary device at a heating rate of 10 °C/min in air up to 650 °C.

### 3.7. Small-Angle X-ray Scattering

SAXS measurements were performed on a laboratory diffractometer (AMUR-K, Institute of Crystallography, Moscow, Russia) at a wavelength *λ* = 0.1542 nm in a Kratky-type (infinitely long slit) geometry covered the range of momentum transfer 0.12 < *s* < 7.0 nm^−1^ (here, *s = 4π sinθ/λ*, where *2θ* is the scattering angle). The scattering profiles were corrected for the background scattering and primarily processed using the program PRIMUS [101] of the software suit ATSAS [102]. The experimental SAXS data were normalized for the intensity of the incident beam, and then a correction for the collimation distortion was made in accordance with the standard procedure [103].

The processed experimental SAXS curves were used to compute the volume size distribution functions *D_V_(R)* of the scattering particles. Assuming the particles to be spherical, an indirect transform program GNOM [95] was employed to solve the integral equation.

### 3.8. Antimicrobial Tests

The antimicrobial activity of Ag NPs/BCF and Ag NPs/SBCB were assessed by the agar diffusion method. Inhibition zones were measured manually using a digital caliper. Assays were performed three times in triplicate. Amphotericin B 40 µg (NII Pasteur, St. Petersburg, Russia) and ampicillin 10 µg (NII Pasteur, St. Petersburg, Russia) were used as positive controls. The antibacterial activity was assessed with the following test strains: Gram-negative bacteria of *Escherichia coli* ATCC 25922; Gram-positive bacteria *Bacillus subtilis* ATCC 6633 and *Staphylococcus aureus* ATCC 25923. The antifungal activity was assessed with mold and yeast fungi: *Aspergillus nige*r INA 00760 and *Candida albican*s ATCC 2091. The test culture of *B. subtilis* ATCC 6633 was grown on Gause 2 medium (g/L): 2.5 tryptone (or 30 mL Hottinger broth), peptone-5, sodium chloride-5 and glucose-10, *Staphylococcus aureus* ATCC 25923 was grown on Müller–Hinton medium and *E. coli* ATCC 25922 was grown on LB (tryptone soy agar). *A. niger* INA 00760 and *C. albican*s ATCC 2091 were grown on PDA (potato dextrose agar).

The turbidity of the bacterial suspension was adjusted to 0.5 McFarland standard (equivalent to 1.5 × 10^8^ CFU/mL) and 1.0 McFarland standard (equivalent to 1.5 × 10^8^ CFU/mL) for fungal strains. The diameter of the inhibition zones was measured after 24 h at 28 °C. The disk diffusion test allows one to evaluate the area at which the active substance stops the bacteria/fungi from growing, which is called the inhibition zone. Comparison of the zones of inhibition allows one to compare the effectiveness of the proposed substances with commercial drugs [104,105]. Assays were performed three times in triplicate.

## 4. Conclusions

Two forms of bacterial cellulose modified with Ag nanoparticles were obtained and studied. SEM data showed that the primary layers of BCF are thinner than those of SBCB, while the secondary layers are thicker in BCF, which characterizes the difference in the rate of layer formation in static and dynamic cultivation. It was determined by TEM that regardless of the BC form, Ag nanoparticles have an average size of 2.8 nm. According to XPS the relative content of C-C/C-H groups in SBCB is 2.6 times that of BCF. This may be assigned to biochemical processes that occur during dynamic cultivation. An increase in the content of C-C/C-H bonds in both forms of cellulose after their modification with silver is associated with the presence of a carbon environment of Ag nanoparticles in the Ag block and their partial transfer to cellulose. At the same time, as in the case of the initial celluloses, the relative content of C-C/C-H is higher on the SBCB surface, which may be due to the difference in their morphologies. The XPS data indicate that Ag^0^ is the predominant state of silver atoms. Positive core-level shift of the Ag 3d_5/2_ peak relative to that of massive silver evidences dispersion of Ag NPs and a particle size of about 3 nm. Both forms show efficacy against *Bacillus subtilis*, *Staphylococcus aureus* and *Escherichia coli* bacteria. Moreover, the results have shown that Ag NPs/SBCB samples obtained with bacterial cellulose produced by dynamic cultivation are more active than those obtained by static cultivation, especially for *Candida albicans* and *Aspergillus niger* fungi. These findings enhance the versatility of the possible practical applications of Ag NPs/SBCB, for example, as agents for disinfecting drinking water or in antiseptic and antimicrobial cover material.

## Figures and Tables

**Figure 1 ijms-24-07667-f001:**
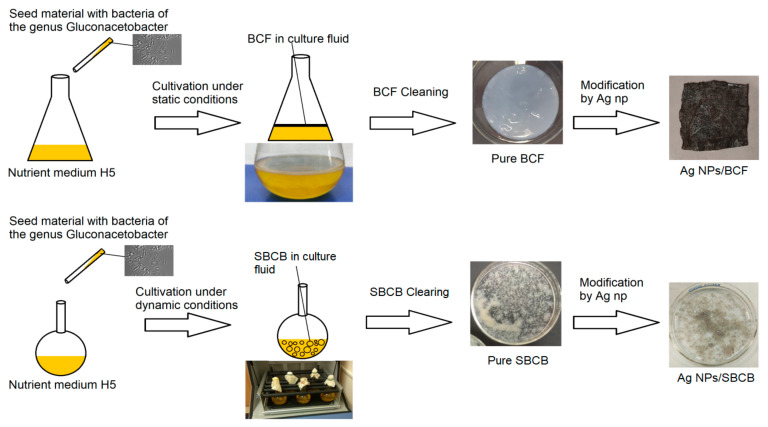
Scheme for obtaining BCF and SBCB with Ag nanoparticles.

**Figure 2 ijms-24-07667-f002:**
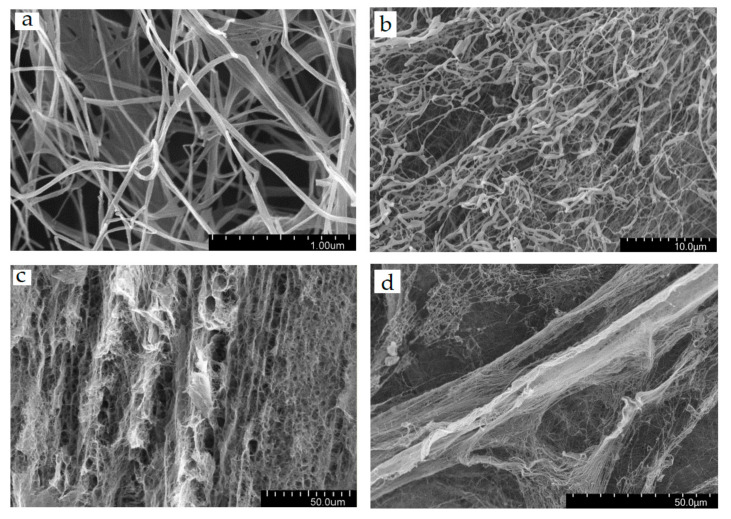
SEM images of the morphology of BCF (**a**,**c**) and SBCB (**b**,**d**) at different magnifications.

**Figure 3 ijms-24-07667-f003:**
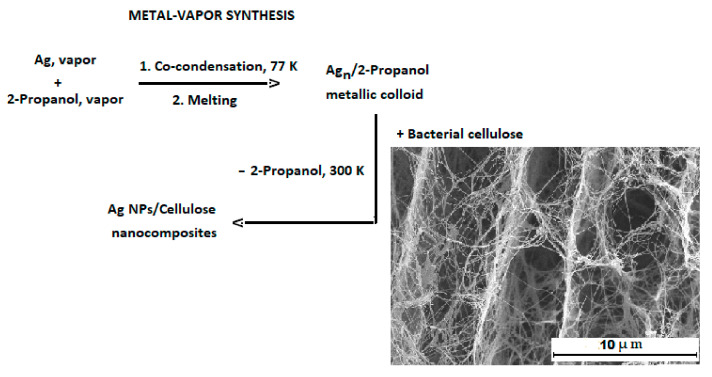
Synthesis of Ag-containing nanocomposites.

**Figure 4 ijms-24-07667-f004:**
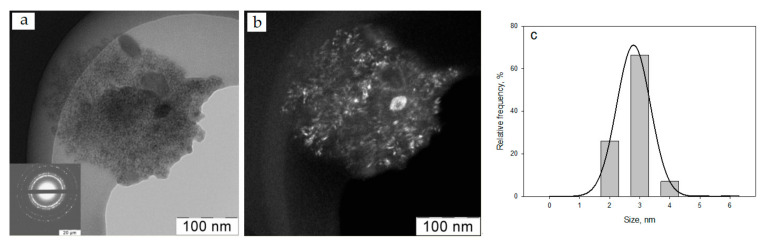
TEM micrographs of Ag nanoparticles on a light background (electron diffraction in the lower left corner) (**a**) and a dark background (**b**); histogram of particle size distribution (**c**).

**Figure 5 ijms-24-07667-f005:**
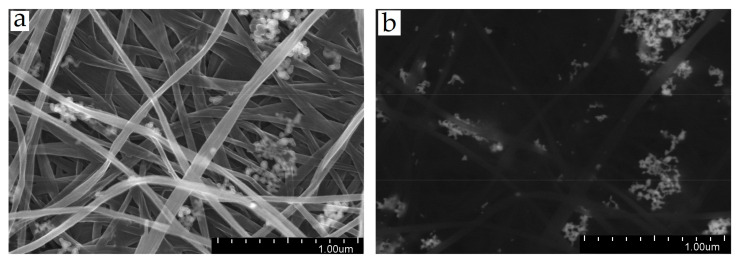
SEM images of the films of BC and composites with Ag NPs: (**a**) Ag NPs/BCF composite; (**b**) Ag NPs/SBCB composite, dark field.

**Figure 6 ijms-24-07667-f006:**
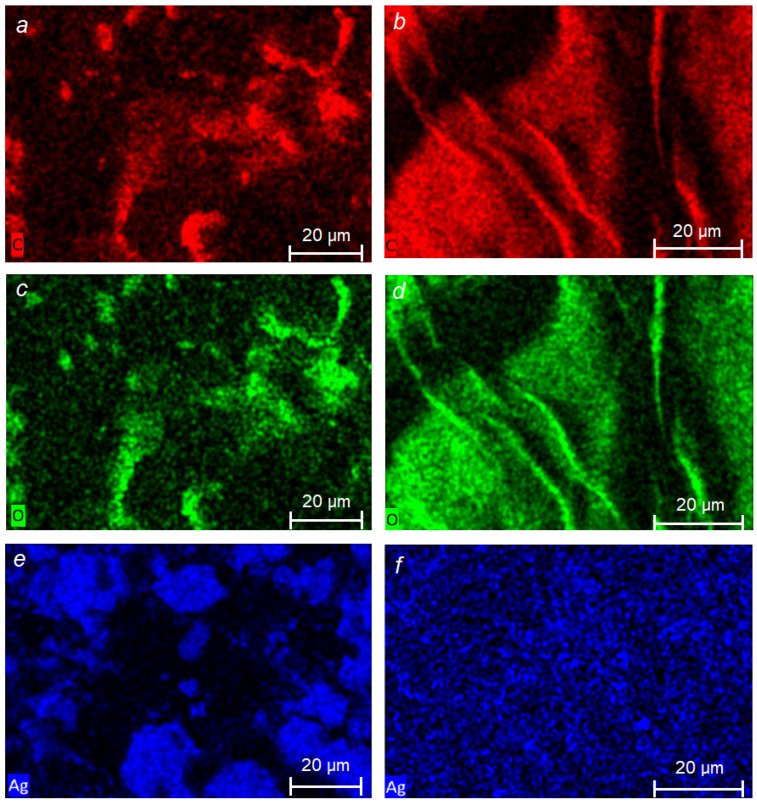
Elemental distribution of C, O, Ag for Ag NPs/BCF (**a**,**c**,**e**) and Ag NPs/BCS (**b**,**d**,**f**) and energy dispersive X-ray spectra of Ag NPs/BCF: C—45.4 at.%; O—39.9 at.%; Si—0.6 at.%; Ag—13.6 at. %; and Al—0.5 at. % (**g**) and Ag NPs/SBCB: C—40.8 at.%; O—46.2 at.%; Si—1.3 at.%; Ag—11.7 at. % (**h**) nanocomposites.

**Figure 7 ijms-24-07667-f007:**
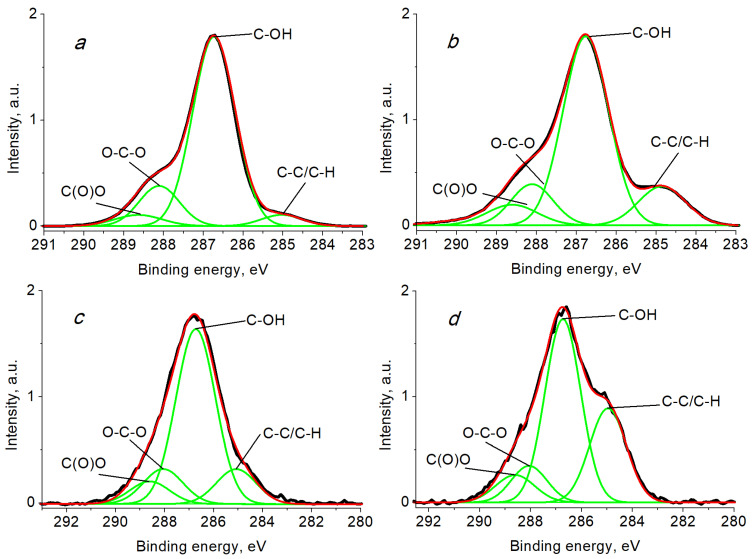
The C 1s photoelectron spectra of samples BCF (**a**), Ag NPs/BCF (**b**), SBCB (**c**), Ag NPs/SBCB (**d**) and Ag NPs (**e**).

**Figure 8 ijms-24-07667-f008:**
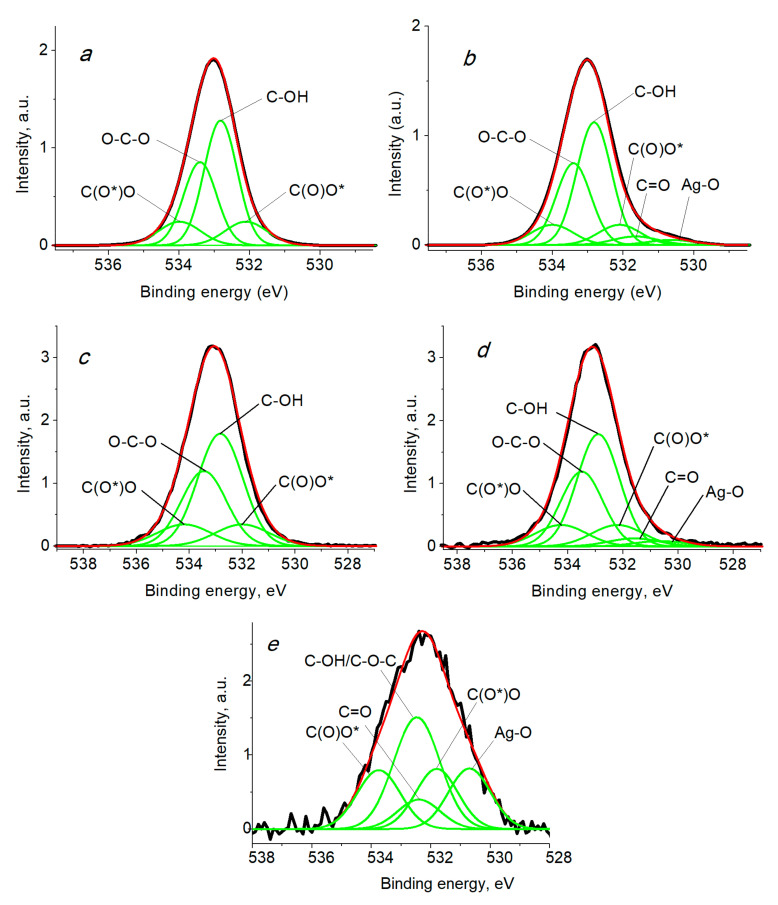
The O 1s photoelectron spectra of samples BCF (**a**), Ag NPs/BCF(**b**), SBCB (**c**), Ag NPs/SBCB (**d**) and Ag NPs (**e**).

**Figure 9 ijms-24-07667-f009:**
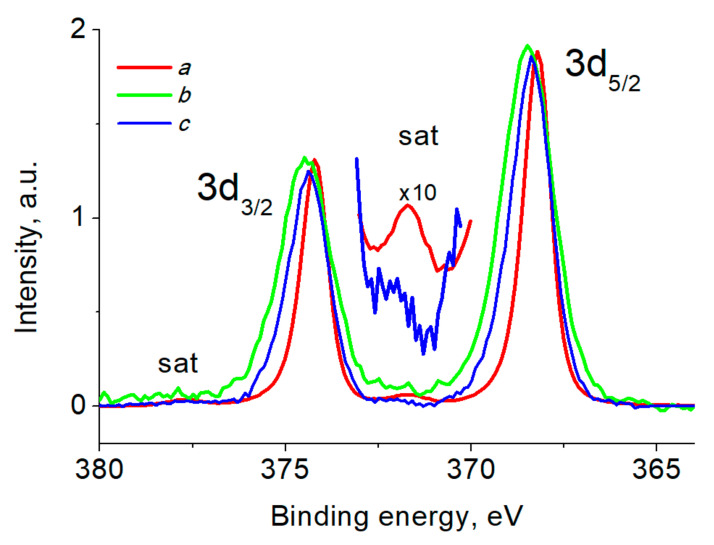
The Ag 3d photoelectron spectra of samples Ag NPs/BCF (a), Ag NPs/SBCB (b) and Ag NPs (c).

**Figure 10 ijms-24-07667-f010:**
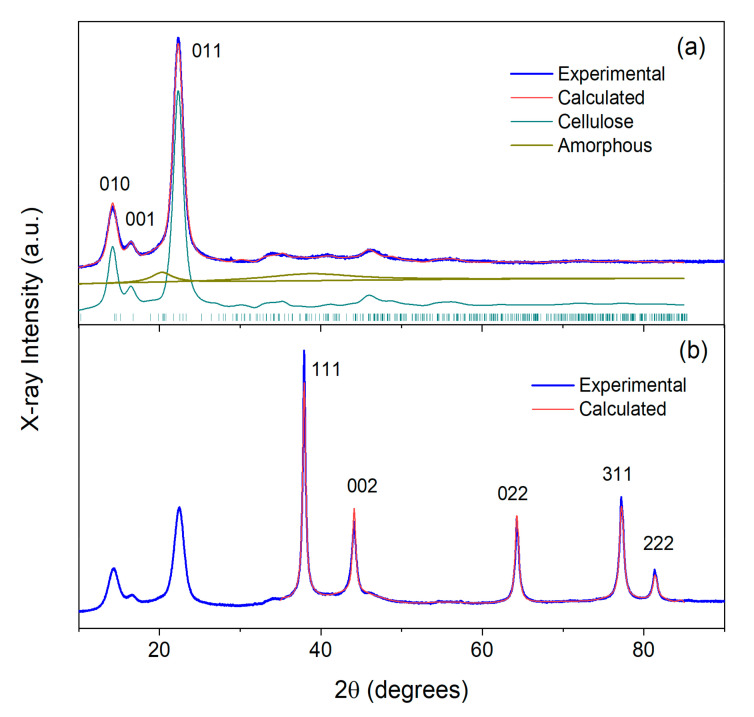
(**a**) X-ray diffraction pattern of initial SBCB (**a**) and composite Ag NPs/SBCB (**b**) and their fits. The main cellulose reflections and reflections of Ag metal are designated by figures.

**Figure 11 ijms-24-07667-f011:**
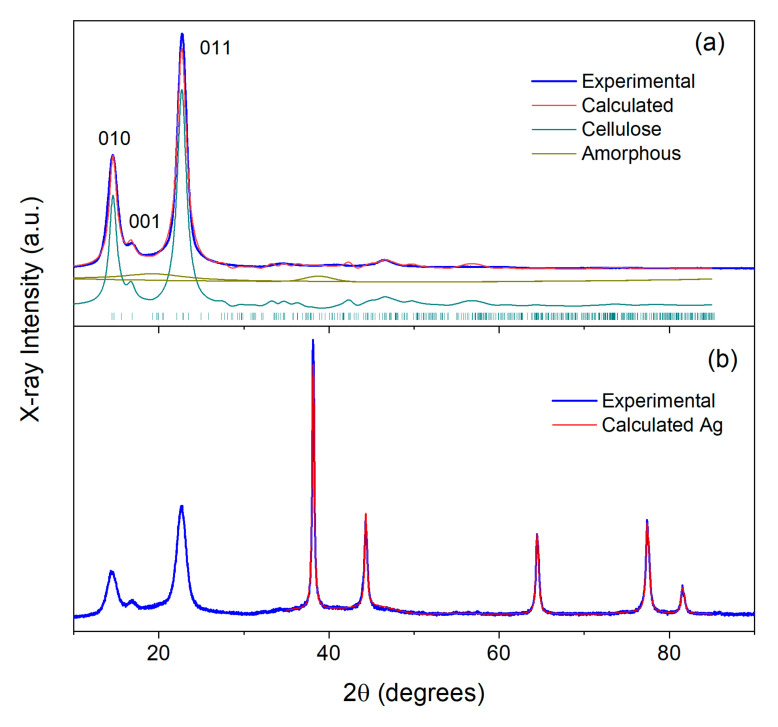
(**a**) X-ray diffraction pattern of initial BCF (**a**) and composite Ag NPs/BCF (**b**) and their fits. The main cellulose reflections and reflections of Ag metal are designated by figures.

**Figure 12 ijms-24-07667-f012:**
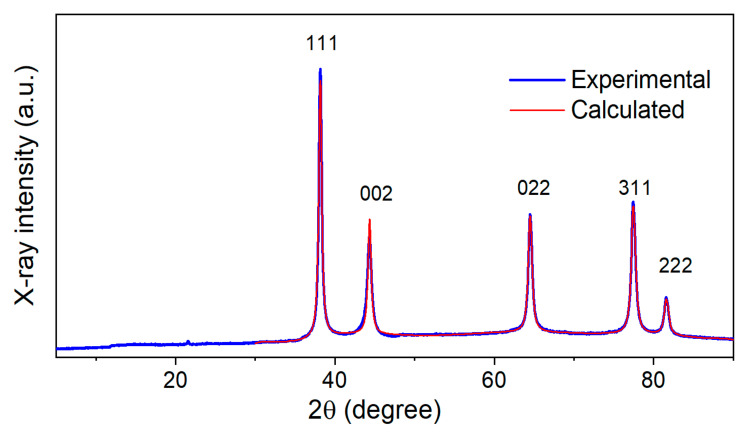
X-ray diffraction pattern of Ag NPs obtained by MVS and its fit.

**Figure 13 ijms-24-07667-f013:**
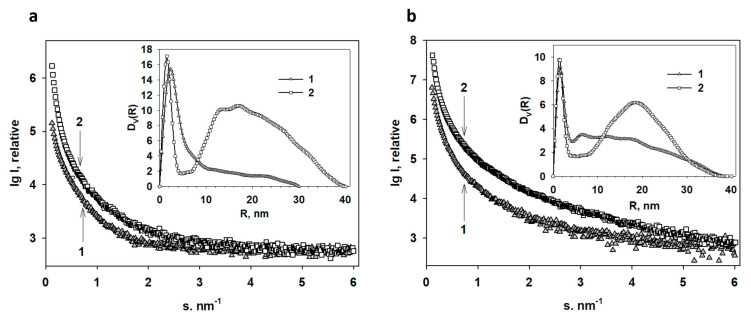
SAXS patterns from the nanocomposites of bacterial cellulose containing Ag nanoparticles: (**a**) experimental SAXS curves from the SBCB gel (1) and Ag NPs/SBCB gel (2). Insert: volume size distribution functions *D_V_(R)* calculated from the curves 1 and 2. (**b**) Experimental SAXS curves from the lyophilized SBCB gel (1) and lyophilized Ag NPs/SBCB gel (2). Insert: volume size distribution functions *D_V_(R)* calculated from the curves 1 and 2.

**Figure 14 ijms-24-07667-f014:**
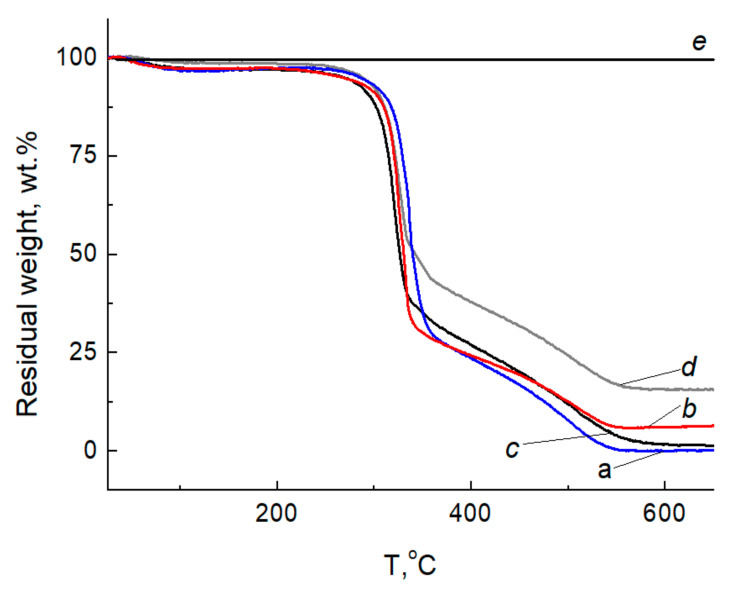
TGA curves BCF (**a**), Ag NPs/BCF (**b**), SBCB (**c**), Ag NPs/SBCB (**d**) and Ag NPs (**e**) at a heating rate of 10 °C/min in air.

**Figure 15 ijms-24-07667-f015:**
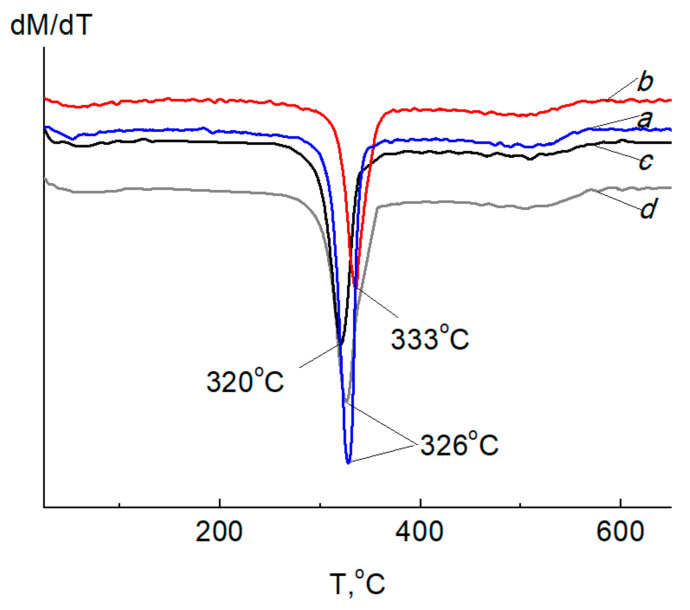
DTG curves of BCF (**a**), Ag NPs/BCF (**b**), SBCB (**c**), Ag NPs/SBCB (**d**) at a heating rate of 10 °C/min in air.

**Table 3 ijms-24-07667-t003:** Characteristics of the C 1s photoelectron spectra: binding energies (E_b_), Gaussian widths (W), and relative intensities (I_rel_) of photoelectron peaks belonging to different chemical groups.

Sample		C-C/C-H	C-OH/C-O-C	O-C-O/C=O	C(O)O
BCF	E_b_, eV	285.03	286.73	288.11	288.63
	W, eV	1.01	1.03	1.03	1.20
	I_rel._	5	75	15	5
SBCB	E_b_, eV	285.08	286.73	288.06	288.61
	W, eV	1.60	1.60	1.60	1.60
	I_rel._	13	65	13	8
Ag NPs/BCF	E_b_, eV	284.88	286.73	288.11	288.63
	W, eV	1.26	1.13	1.03	1.20
	I_rel._	15	65	13	7
Ag NPs/SBCB	E_b_, eV	285.08	286.73	288.06	288.61
	W, eV	1.40	1.40	1.40	1.40
	I_rel._	28	53	11	9
Ag NPs	E_b_, eV	285.0	286.5	288.0	290.1
	W, eV	1.43	1.43	1.43	1.45
	I_rel._	77	13	3	7

**Table 4 ijms-24-07667-t004:** Characteristics of the O 1s photoelectron spectra: binding energies (E_b_), Gaussian widths (W), and relative intensities (I_rel_) of photoelectron peaks belonging to different chemical groups.

Sample		Ag-O	C=O	C(O*)O	C-OH	O-C-O	C(O)O*
BCF	E_b_, eV			532.10	532.82	533.40	534.0
	W, eV			1.4	1.1	1.1	1.4
	I_rel._			0.11	0.47	0.31	0.11
Ag NPs/BCF	E_b_, eV	530.60	531.65	532.10	532.82	533.40	534.0
	W, eV	1.5	1.5	1.4	1.15	1.15	1.4
	I_rel._	0.02	0.04	0.09	0.45	0.30	0.09
SBCB	E_b_, eV			532.0	532.83	533.45	534.20
	W, eV			2.4	2.0	2.0	2.4
	I_rel._			0.11	0.47	0.31	0.11
Ag NPs/SBCB	E_b_, eV	530.65	531.59	532.19	532.89	533.49	534.24
	W, eV	2.2	2.2	2.2	1.8	1.8	2.2
	I_rel._	0.03	0.04	0.1	0.44	0.29	0.1
Ag NPs	E_b_, eV	530.7	532.4	531.8	532.5		533.8
	W, eV	1.42	1.42	1.42	1.55		1.55
	I_rel._	146	71	145	294		145

**Table 5 ijms-24-07667-t005:** Characteristics of the Ag 3d photoelectron spectra: binding energies (E_b_) and spin-orbit splitting (SOS).

Sample	E_b_		SOS	E_b_		State
	Ag 3d_5/2_, eV	Ag 3d_3/2_, eV	Ag 3d_3/2_–Ag 3d_5/2,_ eV	Ag 3d_5/2_ Plasmon, eV	Ag 3d_3/2_ Plasmon, eV	
Ag NPs/BCF	368.19	374.19	6.00	371.66	377.63	Ag^0^
Ag NPs/SBCB	368.47	374.47	6.00	≈372	≈378	Ag^0^, Ag^+^
Ag NPs	368.41	374.40	5.99	372.26	378.06	Ag^0^, Ag^+^

**Table 6 ijms-24-07667-t006:** Antimicrobial activity of Ag NPs/BCF and Ag NPs/SBCB nanocomposites.

Sample	Zone of Inhibition, mm				
	*B. subtilis* ATCC 6633	*S. aureus* ATCC 25923	*E. coli* ATCC 25922	*C. albicans* ATCC 2091	*A. niger* INA 00760
Ag NPs/BCF	7 ± 0.3	7 ± 0.3	8 ± 0.5	7 ± 0.7	8 ± 0.5
Ag NPs/SBCB	9 ± 0.3	9 ± 0.2	10 ± 0.1	10 ± 0.3	10 ± 0.3
Ampicillin 10 µg	29 ± 0.3	24.3 ± 1	30 ± 0.3	nt	nt
Amphotericin B 40 µg	nt	nt	nt	0	23 ± 0.3

nt—not tested.

## Data Availability

Data will be available upon reasonable request from the corresponding author.

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
