# Peer review of "Hybrid Silver-Containing Materials Based on Various Forms of Bacterial Cellulose: Synthesis, Structure, and Biological Activity"

_ijms, 2023, doi:10.3390/ijms24087667_

Round 1

Reviewer 1 Report

Dear Authors

The article entitled “Hybrid silver-containing materials based on various forms of 2 bacterial cellulose: synthesis, structure and biological activity” is a very interesting work. Fallowing are few suggestions which might be helpful for the improvement of the article a bit

1.  The introduction section is very well written and explains the need of the article very nicely. Still the article lacks the actual explanation of the research novelty. It is advisable to give a novelty statement of the research.   

2.  What is the reason for the selection of the two types (Ag NPs/BCF 123 and Ag NPs/ SBCB) of the composition?

3.   Ag itself reported with the antibacterial activity. Authors should compare the zone of inhibition only the silver particles to study the efficacy of the Ag NPs/BCF 123 and Ag NPs/ SBCB.

4. In the conclusion Authors should compare among the Ag NPs/BCF 123 and Ag NPs/ SBCB. Rewrite the conclusion with specific claim. 

A thorough spelling and grammar check is needed specifically for the introduction section. 

Author Response

The article entitled “Hybrid silver-containing materials based on various forms of 2 bacterial cellulose: synthesis, structure and biological activity” is a very interesting work. Fallowing are few suggestions which might be helpful for the improvement of the article a bit/

Response: We thank the reviewer for thorough reading of the manuscript and valuable comments

  1.  The introduction section is very well written and explains the need of the article very nicely. Still the article lacks the actual explanation of the research novelty. It is advisable to give a novelty statement of the research.

Response: We agree with the remark. Additions have been made to the text.

  1. What is the reason for the selection of the two types (Ag NPs/BCF and Ag NPs/ SBCB) of the composition?

Response: We agree with the remark. Additions have been made to the text

  1.   Ag itself reported with the antibacterial activity. Authors should compare the zone of inhibition only the silver particles to study the efficacy of the Ag NPs/BCF and Ag NPs/ SBCB.

Response: In this paper, the authors compare the effectiveness of silver-containing nanocomposites depending on the form of bacterial cellulose.

  1. In the conclusion Authors should compare among the Ag NPs/BCF and Ag NPs/ SBCB. Rewrite the conclusion with specific claim. 

Response: The note taken into account. We have supplemented the conclusions

Reviewer 2 Report

The manuscript by Vasilkov et. al. describing the biological activity of bacterial cellulose- based Ag materials appears to have done with proper synthesis and characterization methods. 
Except the requirement of minor English style editing, this reviewer recommends publishing this article in the present form. 

Author Response

The manuscript by Vasilkov et. al. describing the biological activity of bacterial cellulose- based Ag materials appears to have done with proper synthesis and characterization methods. 
Except the requirement of minor English style editing, this reviewer recommends publishing this article in the present form

Response: We thank the reviewer for thorough reading of the manuscript.  English language of the manuscript has been revised.

Reviewer 3 Report

In this research paper, the authors reported a new nanocomposite based on bacterial cellulose modified by Ag nanoparticles for antimicrobial applications. I recommend resubmission of this work after addressing the following issues.

1.      Please add one more sentence to the beginning of the abstract to state the background and the motivation of this research.

2.      Please add one more sentence to the end of the abstract to put it in a more general context or broader perspective.

3.      Please add one more sentence to the end of the conclusion to put it in a more general context or broader perspective.

4.      Please define bacterial cellulose at the beginning of the first paragraph, such as “a biodegradable and natural cellulose synthesized by bacteria”.

5.      Make sure all scale bars are available and legible for associated figures, such as Figure 2, 3, 5, 6. Delete non-essential information in these figures (e.g., information within the bottom black bars of figure 2).

6.      Consider combining Figure 1 and Figure 3.

7.      I suggest moving the letter labeling of figures (“a, b, c, …”) to the top left position, instead of the current bottom middle position.

8.      The TEM data in Figure 4a are not convincing yet. Do the authors have images that can better represent the particles morphologies in general? I cannot associate Figure 4a and Figure 4b yet.

9.      How did the authors get Figure 4b? By analyzing several (how many?) representative TEM images? What software did the authors use? If yes, please clarify in the main text. Since this technique has been previously used in other research, please add one citation mentioning this method: doi.org/10.1039/C9CC02967B. Also, please include error bars in the figure.

10.  For Figure 7 and Figure 8, avoid using 1-5 for a consistent naming system. Use a-e instead.

11.  The materials and methods section can be moved to Supplementary Materials. Its current position makes the organization of this paper very strange.

12.  A more detailed description of the antimicrobial test is needed, preferably added to the Materials and Methods section. Currently I don’t think the readers can reproduce the results based on the protocol description.

13.  I encourage authors to convert Table 6 to a bar chart.

14.  Figure 16 may be moved to the Supplementary Materials. No apparent and decisive conclusions can be made from looking at these photos with similar appearance. They can be used as supporting evidence for quantitative data.

Author Response

  1. Please add one more sentence to the beginning of the abstract to state the background and the motivation of this research.

Response: We agree with the remark. Additions have been made to the text.

  1. Please add one more sentence to the end of the abstract to put it in a more general context or broader perspective.

Response: We agree with the remark. Additions have been made to the text.

  1. Please add one more sentence to the end of the conclusion to put it in a more general context or broader perspective.

Response: We agree with the remark. Additions have been made to the text.

  1. Please define bacterial cellulose at the beginning of the first paragraph, such as “a biodegradable and natural cellulose synthesized by bacteria”.

Response: We agree with the remark. We added define of bacterial cellulose to the Introduction section

  1. 5.      Make sure all scale bars are available and legible for associated figures, such as Figure 2, 3, 5, 6. Delete non-essential information in these figures (e.g., information within the bottom black bars of figure 2).

Response: We agree with the remark. We have modified figures 2, 3, 5, 6 and deleted non-essential information.

  1. Consider combining Figure 1 and Figure 3.

Response: Figures 1 and 3 show two different processes. We believe that combining them will complicate the understanding of the text.

  1. I suggest moving the letter labeling of figures (“a, b, c, …”) to the top left position, instead of the current bottom middle position.

Response: We agree with the remark.  We modified labeling of figures.

  1. The TEM data in Figure 4a are not convincing yet. Do the authors have images that can better represent the particles morphologies in general? I cannot associate Figure 4a and Figure 4b yet.

Response: We added a TEM image in a dark field and added an addition to the “Materials and methods”

  1. How did the authors get Figure 4b? By analyzing several (how many?) representative TEM images? What software did the authors use? If yes, please clarify in the main text. Since this technique has been previously used in other research, please add one citation mentioning this method: doi.org/10.1039/C9CC02967B. Also, please include error bars in the figure.

Response: We agree with the remark. We have added information in the main text.

  1. For Figure 7 and Figure 8, avoid using 1-5 for a consistent naming system. Use a-e instead.

Response: We agree with the remark.  We have modified labeling of figures.

  1. The materials and methods section can be moved to Supplementary Materials. Its current position makes the organization of this paper very strange.

Response: The article is formed in accordance with the requirements accepted in the journal ( for example, https://doi.org/10.3390/ijms24032318)

  1. A more detailed description of the antimicrobial test is needed, preferably added to the Materials and Methods section. Currently I don’t think the readers can reproduce the results based on the protocol description.

Response: We agree with the remark.  We have added some extra information to the Methods section. The disc diffusion test allows one to evaluate the area at which the active substance stops the bacteria/fungi from growing, which is called the inhibition zone. The comparison of the zones of inhibition allows one to compare the effectiveness of the proposed substances with commercial drugs. Using an in vitro test by the disc diffusion assay, we determined antimicrobial effect of Ag NPs/BCF and Ag NPs/SBCB films.

  1. I encourage authors to convert Table 6 to a bar chart.

Response: We believe that the results in the table look more visual than in the bar chart.

  1. Figure 16 may be moved to the Supplementary Materials. No apparent and decisive conclusions can be made from looking at these photos with similar appearance. They can be used as supporting evidence for quantitative data.

Response: We agree with the remark.   We moved it to the Supllementary Materials.

Reviewer 4 Report

The manuscript presents significant findings that warrant further exploration. There are, however, some comments appended below that should be taken into account as follows:

1. Abstract: The authors should highlight the significant findings and add a conclusion at the end, proposing the potential application of the fabricated materials.

2. Introduction: bacterial cellulose boosted by silver nanoparticles has been previously investigated for biomedical applications; thus, the authors need to elucidate the demand of the suggested materials and novelty of your study.

3. Results and discussion: Fig.6. Scale bars are not visible. For antimicrobial studies: why did you select these microorganisms? Please explain. The potential antimicrobial mechanisms should be explicated according to the structure of the microorganisms. The discussion section should be supported by recent articles. Recent studies demonstrated the toxicity of silver toward cells either with direct influence or indirect effects by overproduction of ROS; thus, the authors should consider expanding their discussion of the limitations of the study. Please check these articles (https://doi.org/10.1016/j.cbi.2022.110166;  https://doi.org/10.1016/j.molliq.2022.121046). It is important to note the potential weaknesses of the study, as this can provide further insight to readers. The antimicrobial activities of the formulated materials are so weak; how did the authors justify the potential application of this material? Did you study the antioxidant capacity of this material using ABTS and DPPH protocols? The authors should study the cytotoxicity of the materials since they intended for biomedical applications.   

4. Methods: Did the authors adjust the bacterial cultures for antimicrobial assays following McFarland 0.5 standard? Please check this article (https://doi.org/10.1016/j.sjbs.2022.01.015) to support this section. The authors should add a section for statistical analysis, illustrating the number of replicates, test and software used in this study.   

Some typos should be corrected.

Author Response

The manuscript presents significant findings that warrant further exploration. There are, however, some comments appended below that should be taken into account as follows:

  1. 1. Abstract: The authors should highlight the significant findings and add a conclusion at the end, proposing the potential application of the fabricated materials.

Response: We agree with the remark. Additions have been made to the text

  1. Introduction: bacterial cellulose boosted by silver nanoparticles has been previously investigated for biomedical applications; thus, the authors need to elucidate the demand of the suggested materials and novelty of your study.

Response: We agree with the remark. Additions have been made to the text

  1. Results and discussion: Fig.6. Scale bars are not visible.

Responce: Changes have been made to Figure 6

For antimicrobial studies: why did you select these microorganisms? Please explain. The potential antimicrobial mechanisms should be explicated according to the structure of the microorganisms. The discussion section should be supported by recent articles. Recent studies demonstrated the toxicity of silver toward cells either with direct influence or indirect effects by overproduction of ROS; thus, the authors should consider expanding their discussion of the limitations of the study. Please check these articles (https://doi.org/10.1016/j.cbi.2022.110166;  https://doi.org/10.1016/j.molliq.2022.121046).  It is important to note the potential weaknesses of the study, as this can provide further insight to readers. The antimicrobial activities of the formulated materials are so weak; how did the authors justify the potential application of this material? Did you study the antioxidant capacity of this material using ABTS and DPPH protocols? The authors should study the cytotoxicity of the materials since they intended for biomedical applications.   

Response: The test microorganisms used in antibacterial and antifungal assay were obtained from Americanl Type Culture collection (ATCC) and stored in the Collection of Microorganisms of Gauze Institute of New Antibiotic. Test microorganisms were taken as a standard set of strains for the study of antimicrobial activity. The potential antimicrobial mechanisms have not been studied in this paper. Literature references containing such information have been added to the text of the manuscript. Currently, in this manuscript we only illustrate the antimicrobial properties of two forms (Ag NPs/BCS - Ag NPs/BCF) to show the perspectives for our further work. At the current stage, the proven antifungal activity of Ag NPs/BCS the assessed by conventional agar diffusion method against C. albicans ATCC 2091 is an interesting finalized result which would become a groundwork for our further fundings.  Also, we have added some extra information in the discussion about our previous cytotoxicity investigations. Cytotoxical effect was studied on mesenchymal stem cell (MSC) primary cell culture from human gums in accordance with ISO 10993-5: 2009 Data indicates the absence of any cytotoxicity against mesenchymal stem cell (https://doi.org/10.1504/IJNT.2019.106615).

  1. Methods: Did the authors adjust the bacterial cultures for antimicrobial assays following McFarland 0.5 standard? Please check this article (https://doi.org/10.1016/j.sjbs.2022.01.015) to support this section. The authors should add a section for statistical analysis, illustrating the number of replicates, test and software used in this study.   

Response: We agree with the remark. We added some extra information to the Methods section.

Reviewer 5 Report

Dear Authors,

The research article looks really good and well explained. I would like to accept for publication.

Author Response

The research article looks really good and well explained. I would like to accept for publication.

Response: We thank the reviewer for thorough reading of the manuscript.

Round 2

Reviewer 1 Report

Dear Authors 

Before submitting, double-check your spelling and grammar. 

Minor Spelling Checking is needed

Author Response

Reviewer 1

Before submitting, double-check your spelling and grammar.

Response: The authors thank the reviewer for the work carried out with our article.

                   The text and English of the manuscript have been improved.

Reviewer 3 Report

I suggest acceptance of this work in current form since the authors have addressed my comments.

Author Response

Reviewer 3.

I suggest acceptance of this work in current form since the authors have addressed my comments.

Response: The authors thank the reviewer for the work carried out with our article.

Reviewer 4 Report

The authors addressed the comments carefully and thoughtfully. They ensured that all feedback was considered and the necessary adjustments were made to the document. 

Author Response

Reviewer 4.

The authors addressed the comments carefully and thoughtfully. They ensured that all feedback was considered and the necessary adjustments were made to the document

Response: The authors thank the reviewer for the work carried out with our article.

Round 3

Reviewer 1 Report

All the best